# Zinc-Containing Sol–Gel Glass Nanoparticles to Deliver Therapeutic Ions

**DOI:** 10.3390/nano12101691

**Published:** 2022-05-16

**Authors:** Prakan Thanasrisuebwong, Julian R. Jones, Salita Eiamboonsert, Nisarat Ruangsawasdi, Bundhit Jirajariyavej, Parichart Naruphontjirakul

**Affiliations:** 1Dental Implant Center, Faculty of Dentistry, Mahidol University, Bangkok 10400, Thailand; prakan.tha@mahidol.ac.th; 2Department of Materials, Imperial College London, South Kensington Campus, London SW7 2AZ, UK; julian.r.jones@imperial.ac.uk; 3Biological Engineering Program, Faculty of Engineering, King Mongkut’s University of Technology Thonburi, Bangkok 10140, Thailand; salita.eia@kmutt.ac.th; 4Department of Pharmacology, Faculty of Dentistry, Mahidol University, Bangkok 10400, Thailand; nisarat.rua@mahidol.ac.th; 5Department of Prosthodontics, Faculty of Dentistry, Mahidol University, Bangkok 10400, Thailand; bundhit.jir@mahidol.ac.th

**Keywords:** bioactive glass nanoparticles, sol–gel process, zinc, osteogenesis

## Abstract

Zn-containing dense monodispersed bioactive glass nanoparticles (Zn-BAGNPs) have been developed to deliver therapeutic inorganic trace elements, including Si, Ca, Sr, and Zn, to the cells through the degradation process, as delivery carriers for stimulating bone regeneration because of their capacity to induce osteogenic differentiation. The sol–gel-derived dense silica nanoparticles (SiO_2_-NPs) were first synthesized using the modified Stöber method, prior to incorporating therapeutic cations through the heat treatment process. The successfully synthesized monodispersed Zn-BAGNPs (diameter of 130 ± 20 nm) were homogeneous in size with spherical morphology. Ca, Sr and Zn were incorporated through the two-step post-functionalization process, with the nominal ZnO ratio between 0 and 2 (0, 0.5, 1.0, 1.5 and 2.0). Zn-BAGNPs have the capacity for continuous degradation and simultaneous ion release in SBF and PBS solutions due to their amorphous structure. Zn-BAGNPs have no in vitro cytotoxicity on the murine pre-osteoblast cell (MC3T3-E1) and periodontal ligament stem cells (PDLSCs), up to a concentration of 250 µg/mL. Zn-BAGNPs also stimulated osteogenic differentiation on PDLSCs treated with particles, after 2 and 3 weeks in culture. Zn-BAGNPs were not toxic to the cells and have the potential to stimulate osteogenic differentiation on PDLSCs. Therefore, Zn-BAGNPs are potential vehicles for therapeutic cation delivery for applications in bone and dental regenerations.

## 1. Introduction

Bone and dental defects are dynamically self-healing. However, the dynamic regeneration in critical bone and dental defects is insufficient for effective healing and regener-ation. Synthetic bone graft substitutes have been clinically used for reconstructive orthopedic surgery and dental implantation to overcome the limitation of using autograft and allograft approaches, such as two surgical sites with donor site morbidity. Bioceramics, including calcium phosphate (CaP), hydroxyapatite (HA), tricalcium phosphate (TCP) and bioactive glasses (BAGs), have been developed in the last few decades because of their unique ability to promote natural bone regeneration and bone-bonding ability [1]. BAGs were clinically used for bone and dental regeneration applications because of their outstanding ability to form strong chemical bonds with surrounding tissues via the hydroxyl carbonate apatite (HCA) layer and their osteogenic, osteoconductive, and osteoinductive potential [2]. Moreover, the biodegradable property and capability of BAGs to deliver the dissolution products have the great potential to stimulate osteogenic marker genes [3,4]. BAGs have been recently developed in nano-sized particles to use as therapeutic carriers because of their ability to enhance the cells’ uptake ability and increase bioactive properties through the release of ions locally inside the cells [5,6,7]. As nanotechnology has been intensively introduced in medical applications, such as using nanoparticles (NPs), including gold nanoparticles, superparamagnetic nanoparticles, lipid-based nanoparticles, polymeric nanoparticles, and silica nanoparticles [8,9,10], in drug delivery systems and imaging because of their unique properties, including a high specific area-to-volume ratio, bioactive glass nanoparticles (BAGNPs) have been introduced to improve the properties of BAGs. BAGNPs have been widely used in biomedical applications because of their small size, large specific surface area, and their large surface-to-volume ratio that gives them special properties [11].

The sol–gel process has been intensely used to prepare bioceramics and bioactive glass [12] due to the low processing temperature, resulting in high purity, homogeneity, and porosity [13]. The sol–gel method consists of hydrolysis and condensation reactions at room temperature, while the traditional melt-quenching process involves melted metal oxide precursors at high temperatures (typically 1200 °C < T < 1550 °C). The sol–gel BAGs were more bioactive than the melt-derived BAGs of the same composition [14]. The binary sol–gel-derived bioactive glass (SiO_2_-CaO) had the ability to form a hydroxycarbonate apatite (HCA) layer [15]. The formation rate depended on the glass composition. The formation rate increased in the BAGs with lower SiO_2_ (50–70% in mol) [16]. The Stöber process was modified to control the size and homogeneity of the particles. Dense monodispersed BAGNPs were commonly synthesized through the sol–gel process, in which the base catalyst, ammonium hydroxide, plays a critical role to control the size and maintain the particles in the solution without gel formation [17]. As the particle size was dominantly controlled by the concentration of the ammonium hydroxide, an increase in the ammonium hydroxide concentration increased the particle size [17,18].

The BAGNPs have had their compositions developed to improve the properties and clinical abilities of traditional BAGNPs over time. Recently, the incorporation of different elements, such as silver (Ag), manganese (Mn), magnesium (Mg), strontium (Sr), zinc (Zn), aluminium (Al), fluorine (F), and potassium (P), into the composition of these BAGNPs to enhance their physical characteristics and therapeutic benefits has been of particular interest [10,19,20,21,22]. Strontium (Sr) is a beneficial trace element in the human body, which represents only 0.035% of the calcium (Ca) content in the skeleton structure [23]. Sr is a divalent cation (Sr^2+^), which closely resembles calcium (Ca^2+^) in its atomic and ionic properties. This biologically beneficial element that is plentiful in human tissues has been reported to promote bone formation because of its capability to stimulate bone formation by activating osteoblasts [24,25,26] and to prevent bone resorption by inhibiting osteoclasts [26,27]. Therefore, Sr has been introduced to sol–gel-derived BAGs to increase new bone formation. It has been reported that Sr-containing BAGs increased the proliferation and alkaline phosphatase activity of osteoblastic cells [28,29]. Zinc (Zn) induced mineralization and bone homeostasis and played a role as a cofactor for enzyme activity and protein synthesis [30]. It also has anti-cancer, antibacterial, and anti-inflammatory properties [10,31,32]. Zn definitely affects chondrocyte and osteoblast functions, whilst inhibiting osteoclast activity [33]. Zn functioned as a network modifier and/or an intermediate oxide in the silica network to modify the BG properties [34]. The ZnO content in glass compositions has affected the biological and physical properties of BAGs. BAGs with low amounts of ZnO (<19 mol%) were amorphous, while those with high amounts of ZnO (>21 mol%) presented as crystalline [35]. The high ZnO content decreased glass degradiation, resulting in reduced bioactivity in simulated body fluid (SBF). An increase in Zn content led to the formation of flake-like structures of calcite and spherical apatite particles, with a much higher ratio of Ca/P, as well as the enhanced formation of bone-bonding calcite and the apatite layer [36].

The objective of this study was to modify the composition of BAGNPs by incorporating Ca, Sr, and Zn using the new two-step post-functionalization process to use as synthetic bone graft substitutes for reconstructive dental surgery and implantation. Our hypothesis is that therapeutic cations released from Zn-BAGNPs could present the synergistic effect on osteogenic differentiation. The murine pre-osteoblast cell (MC3T3-E1) and periodontal ligament stem cells (PDLSCs) were cultured in vitro and the bioreactivity and degradability of zinc-containing BAGNPs (Zn-BAGNPs), following exposure to these cells, were evaluated. The aim was to investigate the biocompatibility, osteogenic differentiation, bioactivity, and degradability of Zn-BAGNPs being developed as an injectable delivery vehicle for bone regeneration.

## 2. Materials and Methods

All reagents were from Sigma-Aldrich (Bangkok, Thailand) unless stated otherwise. The following were used in this study: ethyl alcohol (99.5%), ammonium hydroxide, tetraethyl orthosilicate (TEOS), calcium nitrate tetrahydrate (99%), strontium nitrate (99%), zinc nitrate hexahydrate (≥98%), phosphate-buffered saline (PBS), sodium chloride (NaCl), sodium hydrogen carbonate (Na-HCO_3_), potassium chloride (KCl), dipotassium hydrogen phosphate trihydrate (K_2_HPO_4_·3H_2_O), magnesium chloride hexahydrate (MgCl_2_·6H_2_O), hydrochloric acid (HCl), calcium chloride (CaCl_2_), sodium sulfate (Na_2_SO_4_), nitric acid, minimum essential medium eagle alpha (α-MEM, Gibco^TM^, Bangkok, Thailand), fetal bovine serum (FBS, Thermo Fisher Scientific, Bangkok, Thailand), antibiotic-antimycotic (Thermo Fisher Scientific), trypsin-EDTA (Thermo Fisher Scientific), minimum essential medium eagle alpha modification (α-MEM) with nucleo-sides (Gibco^TM^), sodium bicarbonate, 3-(4,5-dimethylthiazol-2-yl)-2,5-diphenyltetrazolium bromide (MTT, Thermo Fisher Scientific), dimethyl sulfoxide (DMSO), dexamethasone (DEX), β-glycerophosphate, ascorbic acid, paraformaldehyde, RNA isolation kit (Monash total RNA Miniprep), cDNA Synthesis Kit (Bio-Rad, Bangkok, Thailand), iTaq Universal SYBR Green Supermix (Bio-Rad), and Alizarin Red S.

### 2.1. Zn-BAGNP Synthesis

BAGNPs without ZnO were synthesized using the modified Stöber method de-scribed in previous work [18]. The sol–gel-derived dense silica nanoparticles (SiO_2_-NPs) with a diameter range of 100–140 nm were first synthesized. Then 24.7 mL of Milli-Q water, 197.5 mL of ethyl alcohol (99.5%), and 2.9 mL of ammonium hydroxide were mixed in a 500 mL Erlenmeyer flask at a stirring rate of 500 rpm for 15 min. After this, 15.0 mL of tetraethyl orthosilicate (TEOS) was then gently added into the mixed solution and left in the stirrer for at least 8 h in order to complete the hydrolysis and poly-condensation reactions. A white solution was obtained as the hydrolysis and poly-condensation reactions of the silica precursor occurred simultaneously to form the silica network (Si-O-Si) of silica nanoparticles (SiO_2_-NPs). SiO_2_-NPs were collected by centrifugation at 5000 rpm for 40 min and then were simultaneously washed with ethanol (two times) and distilled water (one time) to remove unreacted substances. The SiO_2_-NPs were suspended in Milli-Q water (Figure 1).

After the SiO_2_-NPs were obtained, therapeutic cations including Ca^2+^, Sr^2+^ and Zn^2+^ were then incorporated into the silica network using the post-functionalization process through the one-step and two-step heat treatment processes (Figure 2). For the one-step post functionalization, calcium nitrate tetrahydrate (99%), strontium nitrate (99%), and zinc nitrate hexahydrate (≥98%) were used as the precursors of Ca^2+^, Sr^2+^, and Zn^2+^ and were added to the SiO_2_-NP suspension with a nominal ratio of 1.0 SiO_2_:0.5 CaO:1.5 SrO: x ZnO (where x = 0, 0.5, 1.0, 1.5 and 2.0). This solution was mixed in the ultra-sonication bath for 1 h and was centrifuged at 5000 rpm for 20 min to collect the white pellets of particles. The white pellets were then dried at 60 °C overnight followed by calcination at 680 °C for 3 h at a heating rate of 3 °C/min to obtain zinc-containing BAGNPs (Zn-BAGNPs). The particles were then cleaned with ethanol twice (Figure 2a). For the two-step post functionalization, calcium nitrate tetrahydrate and strontium nitrate were added to the SiO_2_-NP suspension with a nominal ratio of SiO_2_:CaO:SrO of 1.0:0.5:1.5. This so-lution was mixed in the ultra-sonication bath for 1 h and was centrifuged at 5000 rpm for 20 min to collect the white pellets of particles. The supernatant was removed, and then the white pellets were dried at 60 °C overnight. To obtain BAGNPs, Ca^2+^ and Sr^2+^ were incorporated through the calcination at 680 °C for 3 h at a heating rate of 3 °C/min. After that, zinc nitrate hexahydrate was added to BAGNPs with a nominal ratio of SiO_2_:CaO:SrO:x ZnO of 1.0:0.5:1.5:x (where x = 0, 0.5, 1.0, 1.5 and 2.0). These particles were calcined at 550 °C for 3 h at a heating rate of 3 °C/min to form Zn-BAGNPs. The particles were then cleaned with ethanol twice (Figure 2b).

### 2.2. Zn-BAGNP Characterization

Particle size and zeta potential of Zn-BAGNPs were investigated using a nanopar-ticle analyzer (NPA). The surface morphology and particle size of Zn-BAGNPs were imaged using a Transmission Electron Microscope (TEM, JEOL 1400 operated at 120 kV). The dried particles were dissolved in ethanol and were collected on TEM grids. X-ray Diffractometer (XRD) was used to identify the crystallized pattern of the particles. XRD pattern was collected with a Bruker AXS automated powder diffractometer using Cu Kα radiation (1.540600 A°) at 40KV/40mA. Data were collected in the 5–70° 2θ range with a step size of 0.02° and a dwell time of 1.0 s. Fourier transform infrared spectroscopy (FTIR; Thermo Scientific Nicolet iS5) was used in attenuated total reflection (ATR) mode at a wavenumber ranging from 4000 to 400 cm^−1^ at a scan speed 32 scan/min with a resolution of 4 cm^−1^. To determine the elemental composition of each Zn-BAGNPs, X-ray fluorescence (XRF: Fisfer/XUV773) with X-ray generators in a range 8–20 kV operating in a vacuum was used.

### 2.3. Bioactivity Assessment

To compare the release profile of ions from the Zn-BAGNPs with different mol% ZnO, the release of Si, Ca, Sr, and Zn ions from Zn-BAGNPs was monitored as a function of time in two different solutions: simulated body fluid (SBF) at pH 7.4 and phosphate-buffered saline (PBS) at pH 7.4. Kokubo’s method was used to prepare the SBF solution [37]. To prepare 1 L of SBF at pH 7.4, 700 mL of pre-heated (37 ± 1.0 °C) Milli Q water was added into a 1 L polypropylene beaker. The solution was continuously stirred on a hot-plate stirrer to control the temperature at 37 °C. The reagents in Table 1 were gently added in order and stirred until the reagents dissolved clearly. An immediate change in the pH would cause precipitation in the SBF solution [38]. Thus, it is necessary to monitor the pH during the study period. After the reagents were well mixed in the solution, the pH was tuned to 7.40. The prepared solution was adjusted to 1 L with Milli Q water. Before starting the study, the SBF was adjusted to 7.4 at 37 °C. It is noted that the SBF was freshly prepared prior to use. The in vitro bioactivity assessment was conducted by incubating Zn-BAGNPs in SBF solutions at pH 7.4.

### 2.4. Dissolution Study

An amount of 30 mg of Zn-BAGNPs was suspended with 5 mL of media in dialysis tubing that had a molecular weight cut-off of 10 kDa, and immersed into 15 mL of media in 50 mL Falcon^®^ Conical Centrifuge Tubes modified from [19]. All samples were incubated in an incubating shaker at 37 °C shaking at 120 rpm for 1 h, 4 h, 1 day, 7 days, 14 days and 21 days. At each of the time intervals, the pH of the solution was monitored at each specific interval over the period of study, and 0.5 mL of the bulk solution was collected and then immediately replaced with 0.5 mL of the fresh solution. The collected solution was diluted in 2M nitric acid with a 10-fold dilution factor. The ion concentrations of Si, Ca, Sr, and Zn were measured using inductively coupled plasma optical emission spectroscopy (ICP-OES) (Figure 3). At the end of the incubation period, the Zn-BAGNPs were collected by centrifugation and then immediately washed with ethanol and subsequently with acetone to terminate the reactions [38]. The fresh SBF and PBS solutions served as a control.

### 2.5. Cell Culture

MC3T3-E1 cells (ATCC^®^ CRL-2593™) from the murine pre-osteoblast cell line were routinely cultured in T-75 flask under standard condition in a humidified atmosphere at 37 °C and 5% CO_2_ in the basal α-MEM media (Thermo Fisher Scientific), supplemented with 10% fetal bovine serum (FBS, Thermo Fisher Scientific) (*v*/*v*), 100 U/mL antibiotic-antimycotic (Thermo Fisher Scientific). Cells were passaged by trypsinising using trypsin-EDTA (500 μg/mL) (Thermo Fisher Scientific) upon confluence and re-suspended in the α-MEM before cells were counted. The cell stock was diluted to the desired concentration (5 × 10^4^ cells/mL).

Periodontal ligament stem cells (PDLSCs) were obtained from another project after approval by the Institutional Review Board of the Human Ethics Committee of the Faculty of Dentistry, Mahidol University (COA. No. MU-DT/PY-IRB 2017/048.0611). PDLSCs were isolated and characterized in our previous study [39]. The Passages 3–6 of the PDLSCs were used and cultured in basal culture medium containing minimum essential medium eagle alpha modification (α-MEM) with nucleosides (GibcoTM) supplemented with 10% FBS, 2.2 g/L sodium bicarbonate (Sigma-Aldrich), and 100 U/mL antibiotic-antimycotic (Thermo Fisher Scientific) at 37 °C in a humidified 5% CO_2_ atmosphere. The cell stock was diluted to the desired concentration (5 × 10^4^ cells/mL).

### 2.6. Cell Viability Assay

To evaluate the cytotoxicity effect of Zn-BAGNPs, cell viability was measured using MTT colorimetric assay (Thermo Fisher Scientific) according to the manufacturer’s instructions. MC3T3-E1 cells were seeded in flat-bottomed 96-well plates with the cell concentration at 2 × 103 cells/well. The cells were incubated at 37 °C overnight to allow the cell attachment in a monolayer. The medium was removed, and the cells were washed with 100 µL of PBS. The cell culture media was replaced with media containing NPs at concentration range from 0 to 1 mg/mL: 0, 50, 100, 250, 750 and 1000 µg/mL. Cells were exposed to particles for 1 day (direct contact). The control was set as the cells with the medium only. Cell viability was determined using the MTT colorimetric assay based on the conversion of 3-(4,5-dimethylthiazol-2-yl)-2,5-diphenyltetrazolium bromide (MTT) into formazan. The formazan is soluble in dimethyl sulfoxide (DMSO) and the concentration of soluble formazan was determined using a microplate reader (Infinite^®^ 200 Tecan, Port Melbourne, Austria) at 570 nm. The relative cell viability (% viability compared to untreated cells with the particles: control) was calculated as mean value ± standard error of the mean.

PDLSCs were seeded in the flat-bottomed 96-well plates with the cell concentration at 2 × 10^3^ cells/well. The cells were incubated at 37 °C for overnight to allow the cell attachment in a monolayer. The medium was removed, and the cells were washed with 100 µL of PBS once. The cell culture media was replaced with media containing NPs at concentrations ranging from 0 to 1 mg/mL: 0, 10, 50, 100 and 250 µg/mL. Cells were exposed to particles for 1 day and 2 days (direct contact). Cell viability was determined using the MTT colorimetric assay. The formazan is soluble in dimethyl sulfoxide (DMSO) and the concentration of soluble formazan was determined at 570 nm using a microplate reader (Infinite^®^ 200 Tecan, Austria). The relative cell viability (% viability compared to untreated cells with the particles: control) was calculated as mean value ± standard error of the mean.

### 2.7. Osteogenic Differentiation

PDLSCs in the basal medium (α-MEM) were seeded into 24-well culture plates (2 × 10^3^ cells/well). The cells were incubated at 37 °C overnight to allow the cell attachment in a monolayer. The culture medium was subsequently changed into both osteogenic differentiation and basal media. To prepare the osteogenic medium, basal medium (α-MEM) was supplemented with 10 nM dexamethasone (DEX, Sigma-Aldrich), 10 mM β-glycerophosphate (Sigma-Aldrich), and 100 µg/mL ascorbic acid (Sigma-Aldrich). The PDLSCs were treated with Zn-BAGNP at a concentration of 250 µg/mL under both osteogenic differentiation and basal conditions, while the control group comprised only media (without particles). The media were changed twice a week and cultured for 3 weeks.

PDLSCs were fixed with 4% paraformaldehyde in PBS at time intervals of up to 3 weeks in culture (1 week, 2 weeks and 3 weeks). The fixed cells were stained with 2% Alizarin Red S in PBS at pH 4.2 to detect calcified tissue formation. The stained calcium was examined using an inverted microscope (Labomed TCM400).

Real-Time Quantitative Polymerase Chain Reaction (q-PCR) was used to investigate the osteogenic differentiation of PDLSCs. The PDLSCs were seeded in a 6-well plate (at a concentration 2 × 10^4^ cells/mL) and incubated for 1 day. The cells cultured under the basal α-MEM and the osteogenic α-MEM media were used as the controls. After the exposure time, total RNA was extracted using the total RNA isolation kit (Monash total RNA Miniprep) from different treated cells according to the manufacturer’s instructions. The total RNA concentrations were measured using a nanodrop (NANODROP Spectrophotometer). First-strand cDNA was reverse transcribed using a cDNA Synthesis Kit (Bio-Rad) following the manufacturer’s protocol. Then q-PCR was performed using the CFX96™ Real-Time PCR Detection System (Bio-Rad). The q-PCR reactions were prepared using iTaq Universal SYBR Green Supermix (Bio-Rad) following the manufacturer’s protocol. The sequences of the primer pairs in this experiment are shown in Table 2. Then relative gene expression of each gene was calculated by the comparative 2^−ΔΔCt^ method where the target is normalized to the reference gene GAPDH.

### 2.8. Statistical Analyses

Statistical analyses were performed by one-way analysis of variance (ANOVA) in Minitab with the appropriate post hoc comparison test (Tukey’s test). A *p*-value < 0.05 was considered significant. The graphs shown present the results as the mean value with the standard deviation (SD) as the error bars.

## 3. Results

### 3.1. Particle Characterization

The nanoparticles with a diameter of 100–200 nm were synthesized using the modified method described in previous studies [19]. Based on the previous study in [25], a concentration of ammonium hydroxide was used to control the size of the particles. Therefore, 0.28 M TEOS, 6.0 M H_2_O and 0.28 M ammonium hydroxide were used to prepare the SiO_2_-NPs. The successfully synthesized SiO_2_-NPs were homogeneous in size with spherical morphology through the modified Stöber process, which was similar to our previous works [17]. The monodispersed SiO_2_-NPs were first synthesized through the sol–gel process prior to post functionalization with the one-step and two-step processes (Figure 2). In the one-step post functionalization, the cation precursors calcium nitrate, strontium nitrate, and zinc nitrate were added into the SiO_2_-NPs (one pot) at the same time, followed by calcination at 680 °C for 3 h at a heating rate of 3 °C/min. It was found that %mol Ca and Sr in 1.0Zn-BAGNP significantly decreased compared to 0Zn-BAGNP (without Zn) due to the different role of Ca^2+^ and Sr^2+^, acting as a network modifier, and Zn^2+^ acted as both an intermediate oxide and network modifier [22,40]. An interesting point is that %mol of SiO_2_ did not significantly change between particles with and without Zn, as shown in Table 3.

Because of the reduction in %mol Ca and Sr in 1.0Zn-BAGNP compared to that in 0Zn-BAGNP, the one-step post-functionalization process was modified to the two-step post-functionalization process. The monodispersed SiO_2_-NPs were first synthesized prior to Ca and Sr incorporation, at a nominal ratio of 0.5 and 1.5 into the SiO_2_-NPs network, through the first post functionalization step via calcination at 680 °C for 3 h, at a heating rate of 3 °C/min. Afterward, Zn was incorporated into the network through the second post functionalization step via calcination at 550 °C for 3 h, at a heating rate of 3 °C/min to form Zn-BAGNPs, with a nominal ZnO ratio between 0 and 2 (0.2, 0.5, 1.0, 1.5 and 2.0). The temperature at 550 °C was used to eliminate the formation of Zn_2_SiO_4_ [41].

TEM images showed the monodispersed dense spherical nanoparticles within a diameter range of 130 ± 20 nm (Figure 1). Based on previous research, not all of the nominal Ca and Sr added was incorporated into the particles [18]. This study aimed to optimize the amount of Zn in BAGNPs to avoid the formation of small crystalline sodium zinc silicate [36]. The desired nominal ratio of SiO_2_:CaO:SrO:x ZnO was 1.0:0.5:1.5:x (where x = 0, 0.5, 1.0, 1.5 and 2.0). The results from TEM confirmed that there were excess salts on the particle surface at a nominal Si: Zn ratio of 1: 1.5 (1.5Zn-BAGNPs) and 1: 2.0 (2.0Zn-BAGNPs) (Figure 1d,e). This excess salt caused particle agglomeration. Thus, the particles needed to be cleaned with 1 M nitric acid. After the additional washing steps, no excess salt on the particle surface was observed, resulting in the prevention of agglomeration in the particles (Figure 1f–j). TEM images also show that the different Zn ratios did not affect the spherical morphology and particle size.

Elemental mapping was also performed on the 1.0Zn-BAGNPs using AZtec imag-ing software, as shown in Figure 2. Si, Ca, Sr, Zn, and O were detected throughout the entire cross section. XRD patterns of Zn-BGNPs after the normal washing process with ethanol and extra washing process with 1 M nitric acid (two times) and ethanol (two times), then drying at 60 °C for 1 day, are presented (Figure 3). A broad halo between 20 and 25θ for all samples, once the particles were double calcined at 680 °C and 550 °C, indi-cated that calcium oxide (CaO), strontium oxide (SrO), and zinc oxide (ZnO) were incorporated into the amorphous glass structure. Taken together with TEM images in Figure 1, it is clear that no crystalline phases were observed after the extra washing step with 1M nitric acid.

The elemental composition of Zn-BAGNPs after the extra washing process to remove the excess salts on the particle surface using XRF is shown in Table 4. For 0Zn-BAGNPs, with free Zn, the overall amount of network modifiers (Ca^2+^ and Sr^2+^) was the highest compared to others. The amount of Zn increased until a Zn ratio of 1.0, before starting to decrease. These results indicated that there is a cut-off limit of Zn incorporation into the BAGNPs network, corresponding to TEM images of 1.5Zn-BAGNPs and 2.0Zn-BAGNPs, which had the unincorporated salts on the particle surface.

Table 5 shows the textural analysis consisting of surface area, pore volume, and average pore diameter, determined using the Brunauer–Emmet–Teller method (BET). For the average pore diameter, all samples were in a range from 27.76 nm to 35.25 nm, repre-sented as a mesoporous type [42]. The average pore diameter tended to increase from 31.09 nm to 35.25 nm, until it hit a turning point at a Zn ratio of 1.0, before starting to decrease from 35.25 nm to 27.76 nm. According to these results, the Zn incorporation did not significantly affect the pore diameter of the particles. Specific surface areas, ranging from 8.85 to 18.95 m^2^/g, were determined by BET for the obtained particles, as shown in Table 5. An increase in the Zn ratio incorporated in the particles resulted decreasing surface area and pore volume. For the average pore diameter, all samples had an average pore diameter in a range from 27.76 to 35.25, represented as a mesoporous type [36]. The average pore diameter tended to increase (from 31.09 nm to 35.25 nm) when increasing the Zn ratio from 0 to 1.0, before starting to decrease (from 35.25 nm to 27.76). It could be inferred that the average pore diameter did not significantly change when the amount of Zn increased. According to these results, the Zn incorporation did not significantly affect the pore diameter of the particles.

FTIR spectra of all particles exhibited characteristic bands of BAGs absorption bands (Figure 4). Si-O-Si asymmetric stretching was observed at 1050 cm^−1^ [43], while Si-O-Si symmetric stretching was seen at 800 cm^−1^ [44]. Taken together with the XRF results, it could be inferred that incorporated Ca, Sr and Zn did not alter the silica network. Moreover, different amounts of Zn had no effect on bonding within the particles.

### 3.2. In Vitro Release Study

To investigate the dissolution behavior of the particles in the SBF at pH 7.4, an imitated internal environment in a body and PBS at pH 7.4, a salt balance buffer, were used. The particles were immersed in both solutions and were monitored after 1 h, 4 h, 1 day, 4 days, 7 days, 14 days, and 21 days of incubation. After the interval time, the samples were monitored to determine the pH of the solutions. The pH of the SBF and PBS solutions for all samples did not significantly change, as shown in Figure 5. The pH of the solutions was in a range between 7.42 and 7.87 over the immersion period for 21 days at 37 °C, shaking at 120 rpm.

The released Si, Ca, Sr, and Zn from Zn-BAGNPs could be observed through solu-tion-mediated dissolution, as the particles were degraded over the incubation period of up to 3 weeks. An increase in the Si, Ca, Sr, and Zn concentrations was observed with an increase in incubation time, and more Si, Ca, Sr, and Zn ions were released in the SBF solution compared to those in the PBS solution. In the SBF solution (Figure 6a), the release of Si gradually increased with time for a sustained period through the degraded silica network. Ca release increased sharply in the first 24 h, followed by steady release for a sustained period, while Sr release was gradually increased. The Ca and Sr release were similar to previous studies [19]. The increase in Sr concentration released from 0Zn-BAGNPs was two- to three-fold compared to Zn-containing particles in the SBF solution, as Sr was more incorporated inside the particles. Zn release increased with immersion time for 0.5Zn-BAGNPs, 1.0Zn-BAGNPs, and 1.5Zn-BAGNPs. Interestingly, the Zn released from 2.0Zn-BAGNPs rapidly increased after 4 days of immersion in the SBF solution. In the PBS solution (Figure 6b), Si release showed a similar trend to that of Si in the SBF solution, but Si concentration showed lower release in the PBS solution. Ca release remained constant over the immersion period. Sr release gradually increased with immersion time for all particles. Zn release slightly increased in the first 7 days, before levelling off. Most of the samples initially released a few Zn ions after 1-day immersion in both solutions.

The nanoparticles degraded in the SBF solution were confirmed by bright field TEM images (Figure 7a). The morphology of Zn-BAGNPs incubated in the SBF solution changed to a much greater extent compared to that of Zn-BAGNPs incubated in the PBS solution, as the density of the particles was reduced. The reduction in density of the particles indicated that the Si, Ca, Sr, and Zn were released from Zn-BAGNPs. There was little change in the morphology of Zn-BAGNPs incubated in the PBS solution, resulting in the low levels of ions released. Moreover, the particles were aggregated with extra salt on the particle surface (Figure 7b).

### 3.3. In Vitro Cell Viability

The effect of Zn-BAGNPs on cell viability (direct method) was evaluated. The pre-osteoblastic cells (MC3T3-E1) were treated with Zn-BAGNPs for 24 h and untreated cells served as the control. Figure 8a shows that no significant difference in cell viability could be observed among all the groups up to a concentration of 250 µg/mL. Only 0Zn-BAGNPs had no toxicity to the cells up to a concentration of 500 µg/mL. The bright field microscopic images showed the morphology of the treated cells with the particle concentration ranging from 0 to 1000 μg/mL (Figure 8b). An increase in particle concentration resulted in a decrease in the relative cell viability.

The Zn-BAGNPs had no statistically significant difference up to a concentration of 250 μg/mL, following 1-day and 2-day incubation, with the particles directly compared to the unexposed cells, and particle concentration ranging from 0 to 250 μg/mL, indicating the lack of in vitro cytotoxicity for the PDLSCs treated with all Zn-BAGNPs up to a concentration of 250 μg/mL (Figure 9a,b). The fibroblast morphology of PDLSCs (bright field microscopy) after exposure to Zn-BAGNPs for 1 day (Figure 9c) and 2 days (Figure 9d) was detected. It is noted that Zn-BAGNPs did not alter the morphology of PDLSCs, following 1 day and 2 days in culture.

### 3.4. Osteogenic Differentiation

#### 3.4.1. Calcified Formation

PDLSCs were cultured under both basal and osteogenic differentiation media containing the Zn-BAGNPs, at a concentration of 250 µg/mL for 1 week, 2 weeks, and 3 weeks, to evaluate their osteogenic differentiation. At each of the time intervals, differentiated PDLSCs under both media enabled the detection of the calcified formation of bone mineralization, using Alizarin Red S staining, as Alizarin Red S was commonly used to stain calcium deposits in vitro. Under the basal condition, calcium formation was significantly observed after 3 weeks in culture (Figure 10a,c). Under osteogenic conditions, calcium deposits were stained strongly after 2 weeks in culture, in all treated and untreated cells (Figure 10b,d).

#### 3.4.2. Osteogenic Gene Expression Levels

To evaluate the osteogenic properties, the gene expression levels of ALP, Col1a1, Osteonectin, Osterix, and Runx2 of the treated PDLSCs were quantified under both basal (Figure 11) and osteogenic conditions (Figure 12) via qPCR. The expression levels of ALP, Col1a1, and Osteonectin were statistically significantly up-regulated in treated PDLSCs with 0Zn-BAGNPs and 1.0 Zn-BAGNPs after 1, 2, and 3 weeks of culture, both in the presence and absence of osteogenic supplements (* *p* < 0.05). The expression of Runx2 and Osterix in the treated PDLSCs was statistically significantly, having increased after 1 week and 2 weeks under the basal and osteogenic conditions, respectively.

## 4. Discussions

It was previously demonstrated that Sr-containing BAGNPs (Sr-BAGNPs) pro-moted ion release that could potentially enhance mineralizing actions and osteogenic differentiation [24,27]. However, adding Zn to improve the bioactivity of BAGNPs was challenged. Our hypothesis was that Zn-containing dense monodispersed bioactive glass nanoparticles (Zn-BAGNPs) could enhance bone formation due to the released therapeutic ions, including Si, Ca, Sr, and Zn. The aim of this study was to prepare Zn-containing dense monodispersed bioactive glass nanoparticles (Zn-BAGNPs) with a diameter of 130 ± 20 nm through the sol–gel process and two-step post functionalization.

### 4.1. Zn-BAGNPs Characterization

The effect of Zn on the Zn-BAGNPs was investigated. The results indicated that the addition of Zn into Sr-BAGNPs did not affect the size and shape of the particles. The addition of zinc precursors did not affect the morphology and size of particles, while their pore diameter increased in comparison to unloaded Zn (0-ZnBAGNPs). Zn was successfully incorporated into particles though the two-step post-functionalization method. This two-step post-functionalization process did not alter the amount of Ca and Sr in Zn-BAGNPs. This difference is attributed to the fact that whilst Ca^2+^ and Sr^2+^ play a role as network modifiers (ionic bonding) that break the Si-O-Si bonds, forming non-bridging oxygen groups (Si-O-NBO) [40], Zn^2+^ acts partly both as an intermediate oxide (covalent bonding) and network modifier (ionic bonding) [26]. Moreover, a temperature range from 600 °C to 800 °C was reported to generate Zn_2_SiO_4_ formation due to an undesired reaction between ZnO and SiO_2_ [41]. To eliminate the undesired Zn_2_SiO_4_ formation, the calcination temperature was held below 600 °C. Therefore, the Zn was incorporated to form Zn-BAGNPs under the calcination process at 550 °C. Moreover, it has been reported that high amounts of Zn (>21 mol%) led to the formation of small crystalline sodium zinc silicate, resulting in decreasing biodegradation [36]. Therefore, the optimal amount of doped Zn should be considered. The effect of different nominal ratios of Zn, ranging from 0 to 2.0, was investigated. There was excess salt on the particles’ surface at the nominal Si:Zn ratio of 1:1.5 (1.5Zn-BAGNPs) and 1:2.0 (2.0Zn-BAGNPs), indicating that not all added Zn was incorporated into the particles. Thus, it was necessary to remove the unincorporated salts on the particle surface through extra washing steps with 1 M nitric acid (twice) and ethanol (twice). The maximum amount of doped Zn was at a Zn ratio of 1.0, indicating the cut-off limit of Zn incorporation into the BAGNPs network. In terms of bioactive glass, the first melt-derived 45S5 Bioglass^®^ composition consisted of 45 wt% SiO_2_, 24.5 wt% CaO, 24.5 wt% Na_2_O, and 6.0 wt% P_2_O_5_. A higher amount of SiO_2_ resulted in a loss of bioactivity. Therefore, the ratio of silica in the melt-derived bioactive glass was in a range of 45–55 wt% [45]. However, the silica content in the sol–gel-derived bioactive glass was high, up to 55–80% mol, whilst maintaining bioactivity in the bioactive glass [15,46]. In this study, silica contents in all samples were in range of 54–59% mol. These compositions can activate cellular response and stimulate osteogenic differentiation. The elemental mapping result implied that Ca, Sr, and Zn were successfully incorporated into the Zn-BGNPs. There was no diffusion gradient, indicating homogeneity of Ca, Sr, and Zn in the silica network throughout the heat treatment process. Ca, Sr, and Zn functioned as the network modifier and intermediate oxide [22] that generated the amorphous structure inside the particles. Taken together, Zn-BAGNPs could have the potential to perform the sustained release of therapeutic cations from their amorphous network. Moreover, the texture analysis results showed that the samples were highly porous and had large surface area that could increase their benefit of biodegradation and bioactivity, including the ability to form a CaP mineral on the surface [5]. The texture analysis also demonstrated that doping with Zn^2+^ did not significantly change the textural properties of the particles. However, an increase in the Zn ratio incorporated in the particles resulted in a decrease in surface area and pore volume. This could indicate that increasing the amount of Zn reduced the gaps in their particles through the incorporation of Zn ions into the SiO_2_ network. These results are similar to previous research [47].

### 4.2. In Vitro Release Study

The bioactive behavior of the particles was evaluated by detecting the apatite-forming ability, with an immersion test in Kokubo’s simulated body fluid (SBF), compared to in the salt balance buffer (PBS). The results showed that the pH of the SBF and PBS solutions for all samples did not significantly change and was in a range between 7.42 and 7.87 over the immersion period. A large pH change could cause toxicity or another change in the cellular response. Hence, it indicates that none of the samples could be toxic when dissolved in the body fluid. Si, Ca, and Sr ions were released from the particles as a function of incubation time, consistent with previous reports [19]. Zn-BAGNPs were able to release relatively low concentrations in both the SBF and PBS solutions at pH 7.4, whilst maintaining a sustained release of Zn ions up to 21 days. That might be because Zn ions were released in a more acidic environment. It has been reported that Zn release changed in response to the acidic environment due to the acid hydrolysis of Si-O-Zn bonds, corresponding to previous studies [31,35]. At the end of the study period, the morphology change was observed in the Zn-BAGNPs incubated in the solution. The density of the particles incubated in the SBF solution was much more reduced than in Zn-BAGNPs incubated in the PBS solution, confirming that Si, Ca, Sr, and Zn ions were simultaneously released after immersion in the SBF solution. The released Si, Ca, Sr, and Zn ions from the Zn-BAGNPs have great potential for advances in ion therapy for bone repair by delivering therapeutic cations into the bone cells locally. These results corresponded to our previous study, in which strontium-containing bioactive glass nanoparticles, with a diameter range of 100 ± 10 nm, could uptake and localize inside the MC3T3-E1 [19] and hMSCs [24].

### 4.3. In Vitro Cell Viability

The effect of Zn-BAGNPs on MC3T3-E1 and PDLSCs cell viability was investigated. Our experimental results using MTT assay revealed remarkable maintained relative cell viability upon the treatment of MC3T3-E1 and PDLSCs cells with Zn-BAGNPs. When cell viability was decreased to less than 70%, this was used as a cut-off value to evaluate the in vitro cytotoxicity of these particles (ISO 10993-5). The cell viability of untreated cells with Zn-BAGNPs served as the control. Only the undoped Zn particles (0Zn-BAGNPs) did not cause toxicity to the cells up to a concentration of 500 µg/mL, implying that the incorporation of Zn into the BAGNPs had statistically significant effects on the reduction in cell viability. The reason behind this might be because the incorporation of Ca, Sr, and Zn modified the silica network, leading to an acceleration in the degradation of the silicate network, since it is less interconnected [48]. Therefore, the cut-off concentration for the following experiments was at 250 µg/mL for all samples. Taken together with the bright field images, the particle concentration increase led to a decrease in the relative cell viability, indicating that the particle concentration dependently enhanced the cytotoxic effect. Moreover, the Zn-BAGNPs particle concentrations had no toxicity to the PDLSCs up to a concentration of 250 µg/mL, after cells were exposed to the particles for 1 day and 2 days. The morphology of PDLSCs did not significantly change, confirming the biocompatibility of all Zn-BAGNPs. The above results are in agreement with previous reports [47].

### 4.4. Osteogenic Differentiation

BAGNPs have been widely used for the repair or replacement of damaged bone and for dental and periodontal restoration and regeneration because of their osteoinductive and osteoconductive capabilities. It has been reported that BAGNPs could stimulate bone formation [40,45]. Therefore, BAGNPs should be able to stimulate cell proliferation and differentiation. In this study, Alizarin red staining was applied to detect the formation of extracellular matrix mineralization after 7, 14, and 21 days, with and without osteogenic supplements. Calcium formation was detected under the basal condition, following 21 days in culture, and under osteogenic conditions, following 14 days in culture. Cell density and staining intensity of the treated PDLSCs became stronger over the period of study compared to untreated PDLSCs, confirming that the variations in stain intensity for the different PDLSCs were time dependent. Moreover, under osteogenic conditions, cells had high stain intensity, indicating that calcium formation in PDLSCs was retarded when cells were in contact with particles for a while.

Osteogenic gene expression levels, including ALP, Col1a1, Osterix, Osteonectin, and Runx2, were evaluated. Runt-related transcription factor 2 (Runx2) and Osterix are required transcriptional regulators for early and late stages of osteoblast differentiation. ALP and Col1a1 are relative early markers for osteogenic differentiation, expressed during extracellular matrix (ECM) formation [49]. A more rapid increase in the expression levels of osteogenic marker genes was statistically significant, indicating the upregulation of osteoblast differentiation genes. This implies that in the absence of osteogenic supplements, Zn-BAGNPs positively upregulated the osteoblast-specific transcriptional expression factor (Runx2) and played a major role at the early stage of osteoblast differentiation by enhancing the expression of Osterix and osteoblastogenic markers, such as ALP, Col1a1, and Osteonectin. Interestingly, particles with Zn (1.0Zn-BAGNPs) could activate a greater level of gene expression compared to particles without Zn (0Zn-BAGNPs).

## 5. Conclusions

Incorporating zinc through two-step post functionalization into monodispersed strontium-containing bioactive glass nanoparticles (BAGNPs) via the sol–gel process can improve biological properties, whilst maintaining the monodispersity, size within a diameter range of 130 ± 20 nm, and spherical morphology of the particles. Ammonium hydroxide, a base catalyst, played a role in controlling the particle size without the aggregation. The results of elemental mapping using AZtec imaging software coupled with the characteristic bands of BAGs absorption bands, including Si-O-Si asymmetric stretching at 1050 cm^−1^ and Si-O-Si symmetric stretching at 800 cm^−1^, indicating that therapeutic cations, including Ca, Sr, and Zn, were successfully incorporated and well-dispersed into the silica network. A broad halo between 20 and 25θ of all XRD spectra indicated the amorphous structure after cleaning with nitric acid. Si, Ca, Sr, and Zn ions were released up to 223, 281, 22, and 1.7 µg/mL, respectively, in the SBF solution over 21 days. Taken together with the morphological change in the immersed particles, it was implied that Zn-BAGNPs had the ability to show continuous degradation and simultaneous ion release in SBF and PBS solutions due to their amorphous structure. The large pH change could cause adverse change in the cellular response. The released ions did not significantly change the pH of the solutions over the 21-day period. The pH remained in a range between 7.42 and 7.87, suggesting the nontoxicity of Zn-BAGNPs when dissolved in body fluid. The use of Zn-BAGNPs up to a concentration of 250 µg/mL did not cause toxicity to MC3T3-E1 and PDLSCs. Moreover, the significant up-regulating transcriptional regulators (Runx2 and Osterix) and osteoblastogenic markers (ALP, Col1a1 and Osteonectin) indicated that Zn-doped BAGNPs had a positive impact on osteogenic differentiation and the calcified formation compared to undoped particles, in the absence of osteogenic supplements. The osteogenic expression levels of PDLSCs exposed to Zn-BAGNPs, in the presence of osteogenic supplements, suggested a possible synergistic effect. Therefore, Zn-BAGNPs might have great potential for use as synthetic bone graft substitutes for reconstructive dental surgery and implantation.

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
