# Peer review of "Zinc-Containing Sol–Gel Glass Nanoparticles to Deliver Therapeutic Ions"

_nanomaterials, 2022, doi:10.3390/nano12101691_

Round 1

Reviewer 1 Report

Detailed comments:

  1. The English of the text should be checked
  2. The novelty of manuscript should be highlighted more
  3. Please eliminate multiple references. After that, please check the manuscript thoroughly and eliminate ALL the lumps in the manuscript. This should be done by characterizing each reference individually and by mentioning 1 or 2 phrases per reference to show how it is different from the others and why it deserves mentioning. Multiple references are of no use for a reader and can substitute even a kind of plagiarism, as sometimes authors are using them without proper studies of all references used. In the case, each reference should be justified by it is used and at least short assessment provided. 
  4. At Introduction part the authors must be included more information about nanoparticles, sol-gel process in comparison with other processes. Also, must be included more advantages and disadvantage of sol-gel process in comparison with other processes. The following references can be included in the Introduction part to improve the quality of manuscript, because they provide relevant information:
  • The Effect of Different Coupling Agents on Nano-ZnO Materials Obtained via the Sol–Gel Process, Nanomaterials, 7 (12), 2017, 439
  • Zinc Oxide—From Synthesis to Application: A Review. Materials 2014, 7, 2833–2881
  • Preparation and Characterization of Silica Nanoparticles and of Silica-Gentamicin Nanostructured Solution Obtained by Microwave-Assisted Synthesis. Materials, 2021, 14, 2086
  • The influence of new hydrophobic silica nanoparticles on the surface properties of the films obtained from bilayer hybrids. Nanomaterials 2017, 7, 47
  • Physicochemical and Morphological Properties of Hybrid Films Containing Silver-Based Silica Materials Deposited on Glass Substrates. Coatings 2022, 12, 242
  • Sol–gel Synthesis of Zinc Oxide (ZnO) Nanoparticles: Study of Structural and Optical Properties. J. Sci. Islam. Repub. Iran 2015, 26, 281–285
  1. Line 105, All reagents were from Sigma–Aldrich (Thailand) unless stated otherwise. – all reagents must be mentioned
  2. Scheme 1 is unclear; it needs to be redone on a larger scale
  3. Schemes 2 and 3 are unclear, the writing is not understood
  4. Figure 3 is unclear; it needs to be redone on a larger scale. Also, more information about obtained results must be included, comparison between the samples
  5. Lines 51-53, “As nanotechnology has been intensively introduced to medical applications, such as using nanoparticles (NPs) in drug delivery systems ….” – please include the example of nanoparticles
  6. Figures 5, 6, 9, 10 are unclear; it needs to be redone on a larger scale; The legends is not understood either
  7. Figures 11 and 12 are unclear; it needs to be redone on a larger scale;
  8. The Conclusions part must be rewritten, it must contain information about the best results obtained, values, it must not be mentioned Figures
  9. Comparison between the obtained results and measured in this study with other reported studies should be done and included for more clarity (indicate values not just number of reference).
  10. The possible other applications of the prepared membranes must be included
  11. Same Reference are very old [2, 3, 4, 5, 6, 14, 15, 17, 18, 19, 29, 31, 36, 53, 54, 58, 61, 62, 64]. The manuscript must contain the relevant information to be attractive for readers (researchers), because science has advanced, and the information indicated in the manuscript is no longer valid. This part should include observed information, noted in the last 10-12 years.
  12. The References are very many; it is not a review, so References must be limited at maximum 50

Author Response

Dear Reviewer, 

Thank you for your advice on our work. Your comments are very useful for us to improve the manuscript. The manuscript has been revised according to the suggestions and comments (the attached file). 

Best regards,

Reviewer 2 Report

This is a well-organized and well-illustrated research paper, has an important clinical message, and should be of great interest to the readers. The paper focused on developing bioactive zinc containing sol-gel nanoparticles capable of delivering inorganic trace elements which have a scope for application in dental and bone repair. Paragraphing is concise and good, and the article consists of important clinical findings and the paper can be improved in the following aspects.

  1. Is it possible to functionalize multipl einorganic trace elements following the proposed methodogy?
  2. Some research suggests the cytotoxic effects of silica nanoparticles towards cells. What ratio of silica is ideal for use of the porposed formulation in bone and dental therapy.
  3. Figure 8.a. Please use diffferent colors that are contrasting to have a good view of the bar graph. Why is the cell viability low at higher concentration. Was that from resulting silica oe excessive zinc?
  4. Figure 8(b) please make sure there are same number of cells in each section to make the comparison easier.
  5. Figure 9. Why stem cells do not have any cell death at same concentrations used in MTT assay? Any specific reason?

Author Response

Dear Reviewer,

Thank you for your suggestions and comments on our manuscript. Your comments are very useful for us. The manuscript has been revised according to your comments (the attached file)

Best regards,

Round 2

Reviewer 1 Report

Accept